# Projected Changes in Intra-Season Rainfall Characteristics in the Niger River Basin, West Africa

**Uvirkaa Akumaga** [1,*] **and Aondover Tarhule** [2]

[1] Department of Plant and Soil Sciences, Oklahoma State University, 371 Agricultural Hall, Room 272, Stillwater, OK 73019, USA

[2] Department of Geography, Binghamton University, PO Box 6000, Binghamton, NY 13902-6000, USA; atarhule@binghamton.edu

\* Correspondence: uvirkaa.akumaga-1@ou.edu

**Abstract:** The magnitude and timing of seasonal rainfall is vitally important to the health and vitality of key agro-ecological and social-economic systems of the Niger River Basin. Given this unique context, knowledge concerning how climate change is likely to impact future rainfall characteristics and patterns is critically needed for adaptation and mitigation planning. Using nine ensemble bias-corrected climate model projection results under RCP4.5 and RCP8.5 (RCP—Representative Concentration Pathway) emissions scenarios at the mid-future time period, 2021/2025–2050 from the Coordinated Regional Climate Downscaling Experiments (CORDEX) dataset; this study provides a comprehensive analysis of the projected changes in rainfall characteristics in three agro-ecological zones of the Niger River Basin. The results show an increase in the average rainfall of about 5%, 10–20% and 10–15% for the Southern Guinea, Northern Guinea and Sahelian zones, respectively, relative to the baseline, 1981/1985–2005. On the other hand, the change in future rainfall intensities are largely significant and the frequency of rainfall at the low, heavy and extreme rainfall events in the future decrease at most locations in the Niger River Basin. The results also showed an increase in the frequency of moderate rainfall events at all locations in the basin. However, in the Northern Guinea and Sahel locations, there is an increase in the frequency of projected heavy and extreme rainfall events. The results reveal a shift in the future onset/cessation and a shortening of the duration of the rainy season in the basin. Specifically, the mean date of rainfall onset will be delayed by between 10 and 32 days. The mean onset of cessation will also be delayed by between 10 and 21 days. It is posited that the projected rainfall changes pose serious risks for food security of the region and may require changes in the cropping patterns and management.

**Keywords:** cereal yield; climate change; rainfall; rainfall characteristics; Niger Basin; West Africa

## 1. Introduction

The magnitude, timing, and distribution of intra-season or within-season rainfall is vitally important to agro-ecological and social-economic systems in the Niger River Basin of West Africa and, indeed, most of Sub Saharan Africa (SSA) [1–3]. Rainfed agriculture, for example, employs approximately 65% of the labor force, accounts for about 95% of the cultivated area, and contributes between 30% and 70% of the region's Gross Domestic Product (GDP) [3–6]. As a result of such high dependence, deviations from the norm or expected amounts and patterns of rainfall have frequently led to devastating droughts and famines, such as the infamous Sahel droughts of 1970–1973, 1983–1985, and 2011, with tragic loss of lives [1,7,8], social dislocations [7,9–11], and loss of livestock [11,12]. Deviations in rainfall also have adverse impacts on the economies and GDP of the countries of the region [13] and, therefore, the stability of governments [8,11].

Hence, knowledge concerning expected change in future rainfall characteristics and patterns is critically needed for adaptation and mitigation planning [14–17]. Owing to a variety of reasons, however, including data constraints, the majority of studies on West Africa to date have focused on changes in the mean annual rainfall (e.g., [18–24]). Relatively fewer studies have investigated changes in intra-season rainfall characteristics, including, for example, the number, frequency, and intensity of rain events [14–16,25], (Table 1). While total seasonal rainfall is undoubtedly important for various purposes, including water resources management, for other activities, such as crop production, the timing, spacing, and magnitude of the rainy season are more critical [26]. Encouragingly, a number of studies have begun to take advantage of the improving granularity of projected climate data over West Africa to investigate and quantify these dynamics. For example, [15], analyzed statistics for simulated daily rainfall characteristics over West Africa. This study used data produced by 10 Regional Climate Models (RCMs) within the framework of the Coordinated Regional Climate Downscaling Experiments (CORDEX; http://www.cordex.org/). The results showed that while individual RCMs exhibited a wide range of differences associated with higher-order statistics (frequency, intensity of precipitation and extreme daily events), through error cancellation, the multi-model ensemble mean of the indices provided good agreement with the observations. [27], analyzed an ensemble of regional climate projections over the CORDEX African domain, with RegCM4 model driven by the Hadley Centre Global Environment Model (HadGEM) and Max Planck Institute (MPI) global models for the representative concentration pathway (RCP8.5) emission scenario for 1976–2005 and 2070–2099 time periods. Their study focused on the seasonal and intra-seasonal monsoon characteristics, including seasonal totals, onset and cessation and intra-seasonal variability of the monsoon season. They observed a delayed onset and early retreat of the monsoon along with increased intensity of precipitation over the West Africa sub region, implying a shortening of the growing season. Other studies have reached similar conclusions (e.g., [16,22,27–33]). It is worth noting that the projected changes are not spatially homogeneous. Based on a review of more than 200 papers, [17] found that future wet conditions in central-eastern Sahel contrast with dry anomalies over Western Sahel [22,34,35]. They also observed that the increased in rainfall in central-eastern Sahel is linked to a strengthening and northward shift of the meridional overturning circulation over West Africa, reinforcing the monsoonal flow [34]. On the other hand, the projected dry condition in the western Sahel is associated with a reinforcement of the African easterly jet and modifications in the overturning zonal circulation connecting the Indian and Atlantic Oceans. The study concluded that the increase in future rainfall in the central-eastern Sahel region is characterized by a robust increase of the rainfall amounts in September–October (70% of the Coupled Model Intercomparison Project-Phase 5 (CMIP5) model runs; [22]).

In this study, we make use of the CORDEX dataset to further analyze projected changes in intra-season rainfall characteristics in the Niger River Basin (NRB). Given the severity of expected future rainfall changes and their implications, knowledge concerning expected changes at local scale is essential for developing adaptation and mitigation measures within the regional context but with local specificities [33]. Distinct from prior studies, therefore, the present study is site specific, providing finer detail about the risks and changes that stakeholders at the specific locations will have to respond to. Additionally, we investigate these field scale dynamics for three agro-ecological zones, providing a basis for comparison and analysis of spatial differences. We recognize that the results of site-specific analyses are inherently less robust and less spatially representative than regional scale studies. On the other hand, local stakeholders have to respond to changes at the scale at which they operate, not to regional averages, which may be robust in a statistical sense but not necessarily representative of the local scale.

The rest of the paper is organized in the following manner. Section 2 provides a brief description of the study area, conceptual framework, data and methods; results and discussions appear in Section 3; finally, major findings and conclusions are presented in Section 4.

**Table 1.** The conceptual framework.

| Type of Change/Manifestation/Explanation | References | Domain of the Study | Time Period | Models Used | Relevant Findings | Causes of Change |
|---|---|---|---|---|---|---|
| **Scenario 1**. Shift in season either earlier (black curve) or later (red curve), without change in the total amount of rainfall or length of the season.<br><br>An overall season delay in both the start and end of the rainy season in the whole region. Shortening of the rainy season [16,22,26–29,31,36,37] but [38] observed a lengthening of the monsoon season in some locations. | [28] | Sahel | 2075–2099/ 1975–1999 | CMIP3 used, evaluation method not specified. | An overall delay in both the start and end of the rainy season in the whole region. Shortening of the rainy season. | Changes in sea surface temperatures (SST). A phase shift of the annual cycle is a near-global response to GHG forcing and the delay in Sahel rainfall is a manifestation of this response. |
| | [29] | West Africa | 2070–2099/ 1970–1999 | 10 CMIP3 models for special emission scenarios (SRES) A2. Student's t-test is used to evaluate the models | A delayed onset and reduced early rainfall. | Response to global greenhouse gases (GHG) forcings. |
| | [31] | Sahel | 1941–2008, 1960–2000 and 1950–2010 | 17 CMIP3 and 15 CMIP5 models, evaluation method not specified. | Delayed onset of monsoons | Changes in SST, oceanic influences (North Atlantic warming continues to out-pace the global tropical oceans) |
| | [22] | Sahel | 2006–2099/ 1900–2005 | 20 CMIP5 models, evaluation method not specified | Delay in the onset of rainfall. | Response to greenhouse gases emissions (increase in GHGs) and changes in SST |
| | [26] | Burkina Faso, WestAfrica | 2021–2050/ 1971–2000 | 5 regional models (CCLM, HadRM3P, RACMO, RCA, REMO), models evaluated | One week delay in the future onset of the rainy season. Also, delay in cessation. | Changes in SST, oceanic influences (North Atlantic warming continues to out-pace the global tropical oceans) |
| | [27] | Sahel and West Africa | 2070–2099/ 1976–2005 | CORDEX (HadGEM2-ES and MPI-ESM models). The models were evaluated | A forward shift in monsoon season (an overall delay in both the start and end of the rainy season in the whole region) | Changes in SST, oceanic influences |
| | [37] | West Africa | 2080–2099/ 1985–2004 | 3 CMIP5 Models (MPI-ESM-MR, HadGEM2-ES and GFDL-ESM2M). Models were evaluated | A delay of monsoon onset | Result of future greenhouse gases |
| | [16] | West Africa | 2061–2090/ 1961–1990 | 16 CMIP5 Models. No evaluation was mention | A delay in both the start and end of the rainy season in the whole region. Shortening of the rainy season. | Changes in SST, oceanic influences (North Atlantic warming continues to out-pace the global tropical oceans. |
| | [36] | Benin | 2011–2040, 2041–2070 and 2071–2100 relative to 1981–2010 | 3 CORDEX models (MPI-REMO, DMI-HIRAHAM5 and SMHI-RCA4). Models bias- corrected but not evaluated on the parameters investigated | Rainy season projected to start one week early for 2011–2040 and 2041–2070 period but a delay in onset for 2071–2100 period. Cessation will be earlier in all times. The length of the rainy season is both positive (longer) and negative (shorter). | Result of future emissions of greenhouse gases |
| | [38] | Sahel/ West Africa | 2031–2070/ 1960–1999 | 13 CMIP5 models | The onset is delayed in this zone. Also, a delay in the monsoon withdrawal | Changes in SST |

**Table 1.** *Cont.*

| Type of Change/Manifestation/Explanation | References | Domain of the Study | Time Period | Models Used | Relevant Findings | Causes of Change |
|---|---|---|---|---|---|---|
| | | | | **Increase:** | | |
| | [39] | West Africa | 2031–2050/ 1981–2000 | HadCM3, HadRM3P, RCA, ECHAM5, RegCM2, REMO. The models were evaluated | Increase in precipitation over the regions north of the Gulf of Guinea. | Changes in SST |
| | [26] | Burkina Faso, West Africa | 2021–2050/ 1971–2000 | 5 regional models (CCLM, HadRM3P, RACMO, RCA, REMO) and models evaluated. | Increase in mean rainfall | Changes in SST, oceanic influences (North Atlantic warming continues to out-pace the global tropical oceans) |
| | [37,40] | West Africa | 2080–2099/ 1985–20 | 3 CMIP5 Models (MPI-ESM-MR, HadGEM2-ES and GFDL-ESM2M). Models were evaluated. | Increase in rainfall except for the Western Sahel | Result of future greenhouse gases |
| **Scenario 2**. Change in the magnitude of seasonal rainfall either more/increase (blue curve) or less/decrease (red curve) without change in the onset, cessation, or length of the rainy season. | [36] | Benin | 2011–2040, 2041–2070 and 2071–2100 | 3 CORDEX models (MPI-REMO, DMI-HIRAHAM5 and SMHI-RCA4). Models were bias- corrected but not evaluated on the parameters investigated | Increase 0–12% precipitation for 2071–2100 | Result of future greenhouse gases emissions |
| | [41] | Sahel | 2070–2099/ 1900–1999 | 30 CMIP5.The models were evaluated; 7 satisfactorily reproduced past climate | Seven models project 40-300% rainfall increase in Central Sahel over the 21st century. Three models project an increase of over 100% in average rainfall for central and eastern Sahel | Projected changes linked to a combination of local (through radiative forcing changes) and a remote (through tropical SST impacts on atmospheric stability) forcing mechanism |
| | | | | **Decrease:** | | |
| **Increase:** Increase in the annual rainfall in Burkina Faso and West Africa. Increase in annual rainfall for the period 2071–2100 for Benin [26,36,37,39–41]. An overall increase in rainfall intensity (0–15%) [16]. **Decrease:** Decrease in precipitation amount for the period 2011–2040 and 2041–2070 for Benin [36,37,39]. An overall decrease of rainfall frequency (–5%–20%) across the whole West Africa [16,37]. | [39] | West Africa | 2031–2050/ 1981–2000 | HadCM3, HadRM3P, RCA, ECHAM5, RegCM2, REMO. The models were evaluated. | More than 25% decrease in rainfall for all models except RCA. | Changes in SST |
| | [37] | West Africa | 2080–2099/ 1985–20 | 3 CMIP5 Models (MPI-ESM-MR, HadGEM2-ES and GFDL-ESM2M). Models were evaluated | Over the Gulf of Guinea region rainfall intensity decreases during pre- and post-monsoon phases | Result of future greenhouse gases |
| | [36] | Benin | 2011–2040, 2041–2070 and 2071–2100 | 3 CORDEX models (MPI-REMO, DMI-HIRAHAM5 and SMHI-RCA4). Models were bias- corrected but not evaluated on the parameters investigated | Decrease of up to −6% for the period 2011–2040 and 2041–2070 | Result of future greenhouse gases emissions |
| | [16] | West Africa | 2061–2090/ 1961–1990 | 16 CMIP5 Models. Evaluation method not specified | A decrease of rainfall frequency (−5%–20%) across the whole West Africa | Changes in SST, oceanic influences (North Atlantic warming continues to out-pace the global tropical oceans |
| | [27] | West Africa | 2070–2099/ 1976–2005 | CORDEX (HadGEM2-ES and MPI-ESM models). The models were evaluated | Decrease in precipitation | Changes in SST and ocean influences |
| | [42] | Sahel | 2060–2099/ 1960–1999 | 40 CMIP5 Models. Models were evaluated | Decrease of rainfall over the Western Sahel | Changes in SST |
| | [37] | West Africa | 2080–2099/ 1985–20 | 3 CMIP5 Models (MPI-ESM-MR, HadGEM2-ES and GFDL-ESM2M). Models were evaluated | A decrease of rainfall frequency (−5%–20%) across the whole West Africa | Result of future greenhouse gases |

**Table 1.** *Cont.*

| Type of Change/Manifestation/Explanation | References | Domain of the Study | Time Period | Models Used | Relevant Findings | Causes of Change |
|---|---|---|---|---|---|---|
| **Scenario 3**. Change in the magnitude of seasonal rainfall due to later onset of season without change in cessation (green curve), earlier cessation without change in onset (blue curve), or late onset and earlier cessation (red curve).<br><br>A delay in both the start and end of the rainy season in the whole region [16,22,26,28,36]. Other studies projected an earlier cessation [27,28,36,37,39,43]. There is late onset but no change in cessation in some models. For two models, the rainy season period seems to be delayed without any change in the season duration. Thus, the rainy period is not projected to change significantly in these two models despite significant increase in the annual rainfall amount [26,36]. | [28] | Sahel | 2075–2099/ 1975–1999 | CMIP3 used. Evaluation method not specified. | Delay in both the start and end of the rainy season in the whole region | Changes in Changes in SST and GHG forcings |
| | [22] | Sahel | 2006–2099/ 1900–2005 | 20 CMIP5 models. Models were evaluated | Delay in both the start and end of the rainy season in the whole region | Result of future greenhouse gases emissions (increase in GHGs) and changes in SST |
| | [26] | Burkina Faso, West Africa | 2021–2050/ 1971–2000 | 5 regional models (CCLM, HadRM3P, RACMO, RCA, REMO). Models were evaluated | Delay in both the start and end of the rainy season in the whole region | Changes in SST, oceanic influences (North Atlantic warming) |
| | [16] | West Africa | 2061–2090/ 1961–1990 | 16 CMIP5 Models. Evaluation method not specified | Delay in both the start and end of the rainy season in the whole region | Changes in SST, oceanic influences (North Atlantic) |
| | [36] | Benin, West Africa | 2011–2040, 2041–2070 and 2071–2100 | 3 CORDEX models (MPI-REMO, DMI-HIRAHAM5 and SMHI-RCA4) | Delay in both the start and end of the rainy season in the whole region for 2071–2100 period | Result of future greenhouse gases emissions |
| | [28] | Sahel | 2075–2099/ 1975–1999 | CMIP3 used. Evaluation method not specified | Late onset and early cessation | Changes in Changes in SST and GHG forcings |
| | [39] | West Africa | 2031–2050/ 1981–2000 | HadCM3, HadRM3P, RCA, ECHAM5, RegCM2, REMO. Models were evaluated. | Late onset and early cessation | Changes in SST |
| | [43] | West Africa | 2070–2099/ 1970–1999 | 10 CMIP3 models for SRES A2. Student's t-test is used to evaluate the models | Late onset and early cessation | Response to GHG forcings |
| | [27] | West Africa | 2070–2099/ 1976–2005 | CORDEX (HadGEM2-ES and MPI-ESM models). The models were evaluated | Late onset and early cessation | Changes in SST and ocean influences |
| | [37] | West Africa | 2080–2099/ 1985–20 | 3 CMIP5 Models (MPI-ESM-MR, HadGEM2-ES and GFDL-ESM2M). Models were evaluated. | Late onset and early cessation | Result of future greenhouse gases emissions |
| | [36] | Benin | 2011–2040, 2041–2070 and 2071–2100 | 3 CORDEX models (MPI-REMO, DMI-HIRAHAM5 and SMHI-RCA4). Models were- bias-corrected but not evaluated on the parameters investigated | Late onset and early cessation | Result of future greenhouse gases emissions |
| | [26] | Burkina Faso, West Africa | 2021–2050/ 1971–2000 | 5 regional models (CCLM, HadRM3P, RACMO, RCA, REMO). Models were evaluated | Late onset, no change in cessation. | Changes in SST, oceanic influences (North Atlantic warming continues to out-pace the global Tropical oceans) |
| | [36] | | 2011–2040, 2041–2070 and 2071–2100 | 3 CORDEX models (MPI-REMO, DMI-HIRAHAM5 and SMHI-RCA4). Models were bias-corrected but not evaluated on the parameters investigated | Late onset but no change in cessation. | Result of future greenhouse gases emissions |

**Table 1.** *Cont.*

| Type of Change/Manifestation/Explanation | References | Domain of the Study | Time Period | Models Used | Relevant Findings | Causes of Change |
|---|---|---|---|---|---|---|
| **Scenario 4** Change in the within season distribution of rainfall from historical (black curve) to a right skewed distribution (green curve) or a left skewed distribution (red curve).  The projected precipitation increase in the central-eastern Sahel is characterized by a robust increase of the rainfall amounts in September–October [22,38]. There is a redistribution of precipitation in monsoon regions. Early summer decreases and late summer increases in precipitation are evident in the African regions [43]. Detailed regional analyses of CMIP5 experiments indicate a redistribution of rainfall within the rainy season in West Africa [22,43]. Rainfall anomalies are predominantly negative at the beginning of the rainy season (May and June), but positive at its end (October), indicating a delay of the main rainy season [28,38]. | [22] | Sahel | 2006–2099/ 1900–2005 | 20 CMIP5 models. Models are evaluated | Increased rainfall amounts in September–October (an indication of left skewed seasonal rainfall over the Sahel). | Response to greenhouse gases emissions (increase in GHGs) and changes in SST |
| | [38] | Sahel/ West Africa | 2031–2070/ 1960–1999 | 13 CMIP5 models | Increased rainfall amounts in September–October (an indication of left skewed seasonal rainfall over the Sahel). | Changes in SST |
| | [43] | West Africa | 2070–2099/ 1970–1999 | 10 CMIP3 models for SRES A2. Student's t-test is used to evaluate the models | Less rain at the start of the season but more rain at the end of the season (an indication of the right skewed seasonal rainfall) | |

## 2. Study Area, Data and Method

### 2.1. Study Area

This study is focused on six locations within the Niger River Basin (Figure 1), distributed in three agro-ecological zones, namely; semi-arid (Sahelian) zone (represented by study sites at Dori, Tahoua and Tillabery); the southern Guinea zone (represented by Makurdi) and the northern Guinea zones (represented by Samaru and N'Tarla). Within the semi-arid Sahel, mean annual rainfall declines from 750 mm in the south to 250 mm in the northern limit of the zone, concentrated in a single-peaked rainy season spanning four to five months (i.e., May/June to September/October). The zonal mean annual rainfall is about 500 mm (1981–2010), allowing a growing period of 90 to 120 days. Agricultural production is dominated by cereals and livestock nomadic herding. To the south of the Sahel is the northern Guinea zone, which also experiences a unimodal rainfall season. In this zone, the annual rainfall declines from 1400 mm in the south to 750 mm in its northern limit with zonal mean average of 1050 mm (1981–2011). The rainy season is longer (five to six months), allowing a growing period of 150 to 180 days. Major crop types include maize, sorghum, millet, rice, yams, groundnut, soybean and beans. In the southern Guinea zone, rainfall reaches 1600 mm distributed across seven months, allowing a growing period of 150 to 210 days. Most common types of crops include grains, notably maize and rice, as well as root crops, including yams and cassava.

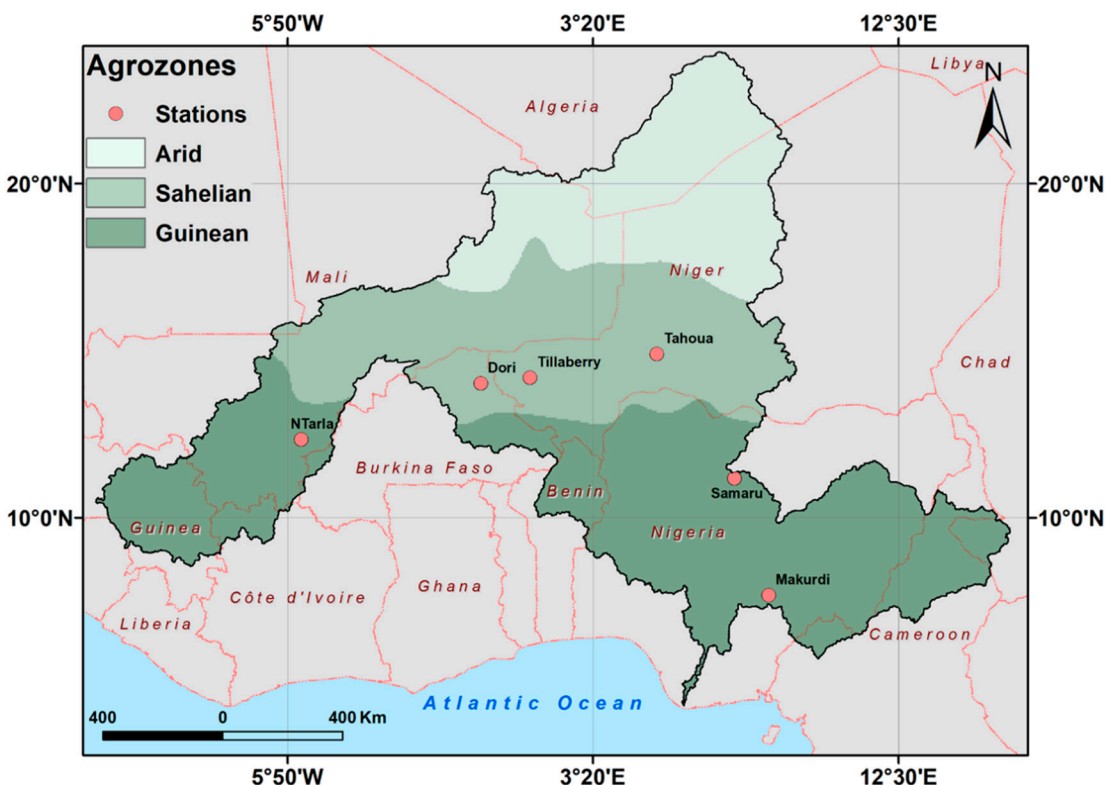

**Figure 1.** The Location of the Niger Basin in West Africa, showing study locations (red dots) and agro-ecological zones.

### 2.2. Conceptual Framework and Data

Table 1 conceptually illustrates several scenarios depicting how a location might experience changes in intra-season rainfall. Each scenario has implications for different sectors and activities. For example, a seasonal rainfall shift (scenario 1) will necessitate changes in traditional sowing and harvesting dates, possibly conflicting with, or displacing the timing of, other non-agricultural activities during the year. Similarly, a delay in the onset and early cessation (scenarios 3) will shorten the

growing period, increasing the risk of crop failures or reduced yields, especially for long duration cultivars. The apparent simplistic and orderly scenarios shown in Table 1 is for purposes of illustration and clarity of presentation only. In practice, rainfall changes may involve complex combinations of several scenarios contemporaneously.

### 2.3. Data

Historical daily rainfall data were obtained from the national meteorological agencies of Nigeria (for Makurdi and Samaru), Burkina Faso (Dori), Republic of Niger (Tahoua and Tillabery) and Mali (N'Tarla). The data come from principal stations, which maintain World Meteorological Organization (WMO)-approved standards of data collection procedures and quality control. As such, no further quality control was implemented. For projected rainfall, we relied on data from CORDEX, Africa [44]. The datasets are available at $0.5 \times 0.5$ degree resolution for West Africa. Projected daily rainfall data were obtained for the six study locations for the period 2021–2050 for two representative concentration pathways (RCP4.5 and RCP8.5). The reference period for observational data is 1976–2005. At two sites, however, observational data were available only for 1980–2005, necessitating adjustment of the projected data used to 2025–2050. All of the data were simulated by nine global climate models (GCMs) and one regional climate model (RCM). The GCMs/RCM include: CCCma-CanESM2/RCA4; CNRM-CERFACS/RCA4; CSIRO-Mk3-6-0/RCA4; IPSL-CM5A/RCA4; MIROC-MIROC5/RCA4; HadGEM2-ES/RCA4; MPI-ESM/RCA4; GFDL-ESM2M/RCA4; ICHEC-EC-EARTH/RCA4. The selected models have all been shown to have skill in reproducing the current mean climatology (Table 2) [21,24,45,46] and key features of the present-day precipitation over West Africa (e.g., [14,15,25,27]).

Bias correction followed the methodology described by [47]. In summary: from the CORDEX dataset, data for the nearest model grid to the study location were extracted and used for the bias-correction process. This approach has been shown to be superior to averaging multiple grid point time series which degrades the statistics, in particular at the high intensity end of the distribution [48]. The bias correction was done separately for each month. That is, all daily values corresponding to a given calendar month, for the observed and the historical simulation, over the observational period are collected in two time series of equal length. For example, all the 31 daily precipitation values for the month of January from 1976 to, and including 2005, are used to calculate the January bias correction parameters, as explained by [47], (pp. 4–6). The observed and simulated time series are sorted in order of increasing intensity and then plotted against each other. The resulting plot is called an emerging or perfect transform function (PDF). The majority of events cluster near the origin of the plot, because most rainfall time series are dominated by zeros i.e. non rainy days. Consequently, it is customary to ignore this portion of the plot and to fit the remaining data points with the appropriate analytic function [49]. For this study, the plot was satisfactorily approximated by a first degree polynomial as seen in the supplementary material [Figure S3]. The nine bias-corrected cumulative distribution functions (CDFs) (green lines) are almost perfectly superimposed onto the observed cumulative distribution function CDF (blue line), while the non-bias-corrected CDFs (red lines) are spread out, showing that the PDF is well approximated by a linear fit. This fitted TF was then used to correct projections of future scenario precipitation values.

**Table 2.** Evaluation summary statistics for average annual rainfall, onset, cessation, duration and rainfall intensity of the growing season, 1976/1980–2005.

| Agro-Zone | Location | F-test for Variance | | | | | | | | t-test for Difference of Means | | | | | | | |
|---|---|---|---|---|---|---|---|---|---|---|---|---|---|---|---|---|---|
| | | Ann. | Onset | End | Dur. | Rainfall Intensity | | | | Ann. | Onset | End | Dur. | Rainfall Intensity | | | |
| | | | | | | Low | Mod. | Heavy | Ext. | | | | | Low | Mod. | Heavy | Ext. |
| Southern Guinea | Makurdi | 0.001 | 0.069 | 0.000 | 0.006 | 0.498 | 0.490 | 0.478 | 0.023 | 0.978 | 0.101 | 0.776 | 0.724 | 0.856 | 0.950 | 0.840 | 0.230 |
| Northern Guinea | Samaru | 0.001 | 0.005 | 0.031 | 0.049 | 0.485 | 0.473 | 0.496 | 0.158 | 0.597 | 0.839 | 0.060 | 0.672 | 0.851 | 0.889 | 0.918 | 0.745 |
| Sahel | Tahoua | 0.001 | 0.001 | 0.278 | 0.004 | 0.489 | 0.469 | 0.448 | 0.135 | 0.944 | 0.877 | 0.110 | 0.127 | 0.994 | 0.936 | 0.913 | 0.683 |
| | Dori | 0.001 | 0.001 | 0.313 | 0.000 | 0.448 | 0.469 | 0.483 | 0.043 | 0.646 | 0.127 | 0.337 | 0.655 | 0.755 | 0.944 | 0.864 | 0.264 |

Note, Ann. = Annual, Dur. = Duration, Mod. = Moderate, Ext. = Extreme.

*2.4. Methods*

GCMs traditionally simulate changes in the mean (especially ensemble mean) more robustly than variance [21]. Therefore, we utilized the *t*-test for difference of means to evaluate how well the simulated model ensembles reproduced the means of observed rainfall characteristics, recognizing the limitation of comparing regional and grid level data.

Prior to applying the *t*-test, the time series of simulated and observed historical data were first subjected to an *F*-test for variance. Where the variances are not statistically different ($\alpha = 0.2$) we tested for the difference in means assuming equal variance. Conversely, where the *F*-test is significantly different, we used the *t*-test assuming unequal variance. Then we analyzed the projected changes in rainfall characteristics for the target period relative to baseline.

*2.5. Onset, Cessation and Duration of the Rainy Season*

Researchers have employed numerous different criteria to define the agriculturally meaningful onset, cessation and duration of the rainy season in West Africa (see, e.g., [50–56]. For this study, we adopted the criteria of [52,53] based on the recommendation of AGRHYMET (2000). Accordingly, the date of onset is defined as that date from 1 January onward, when rainfall accumulated over a maximum of three consecutive days is at least 20 mm, and when no dry spell within the next 30 days exceeds 10 days. The date of cessation of rains is taken as that date after September 1 following which no rain occurs over a period of 20 days. The duration of the rainy season is the difference between cessation and the onset of rains.

*2.6. Daily Rainfall Frequency and Intensity Analysis*

To investigate possible change in daily rainfall intensity, four intensity categories were prescribed and analyzed, namely light rainfall (<10 mm/day), moderate (10.1 mm–25 mm), heavy (25.1 mm–65 mm) and extreme (>65 mm). These categories have previously been shown to be meaningful for crop production in West Africa (see [50,57]. For each category, we tested for differences in the amount and frequency of rain events in the CORDEX projected and observational data using box plots and the *t*-test for means.

For each study location, the ensemble time series of the total rainfall, onset, cessation, duration, frequency and intensity were derived for the future time period. The ensemble means were obtained by taking the average of the nine climate models after calculating the indices for each of the individual climate models. To determine changes in the projected rainfall, the projected rainfall characteristics were compared with the baseline conditions using box and whisker plots.

## 3. Results and Discussion

*3.1. Evaluation of the Simulated Rainfall for the Historical Period*

Table 2 summarizes results of the *F*-test for variance and *t*-test for means (assuming unequal variance) for each pair of observational and simulated rainfall characteristic for the historical period. Figures 2–4 shows illustrative box and whisker plots of the comparison of the CORDEX multi-models ensemble and observed intra-season rainfall variables at the study locations. For reasons of space, the complete set of evaluation plots appear as supplementary figures [Figures S1–S2].

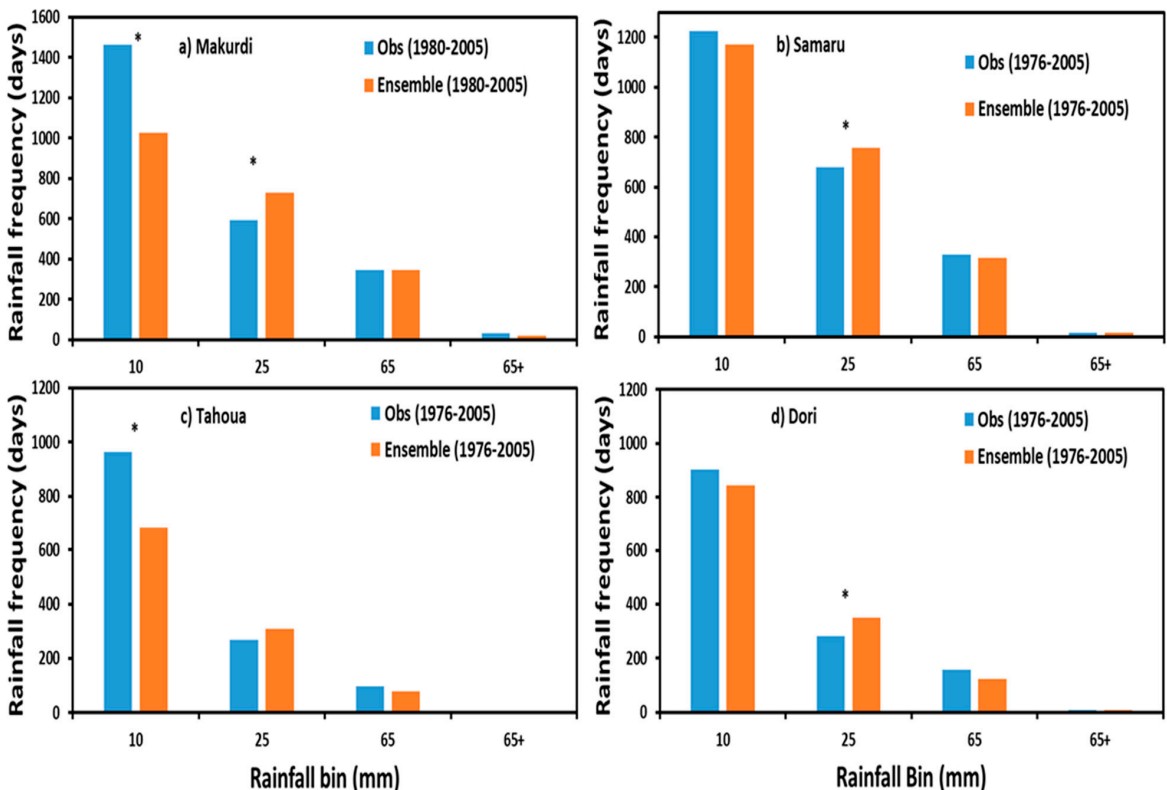

**Figure 2.** A comparison of the Coordinated Regional Climate Downscaling Experiments (CORDEX) multi-models ensemble and observed frequency of rainfall at different intensities (low, moderate, heavy and extreme) over the (**a**) Southern (Makurdi) and (**b**) Northern (Samaru) Guinea agro-ecological zones and (**c**,**d**) the Sahelian (Tahoua and Dori) agro-ecological zones for the historical period, 1976/1980–2005. Note, * indicate where the events are significantly different at 0.05 using *t*-test for mean.

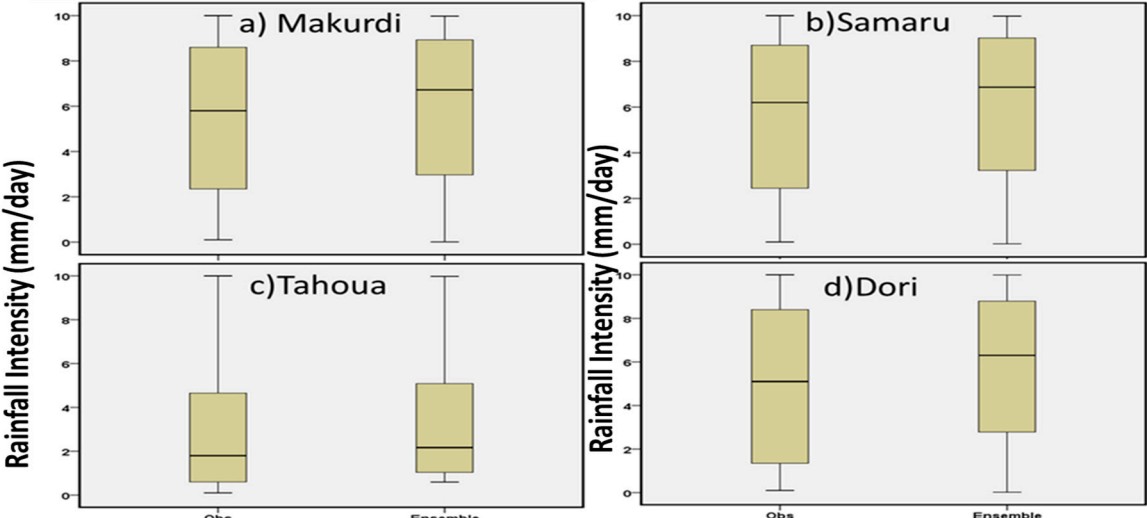

**Figure 3.** Box and whisker plot comparing the CORDEX multi-models ensemble and observed mean daily low rainfall intensity (>0 to ≤10 mm) over the (**a**) Southern (Makurdi) and (**b**)Northern (Samaru) Guinea agro-ecological zones and the (**c**,**d**) Sahelian (Tahoua and Dori) agro-ecological zones for the historical period, 1976/1980–2005.

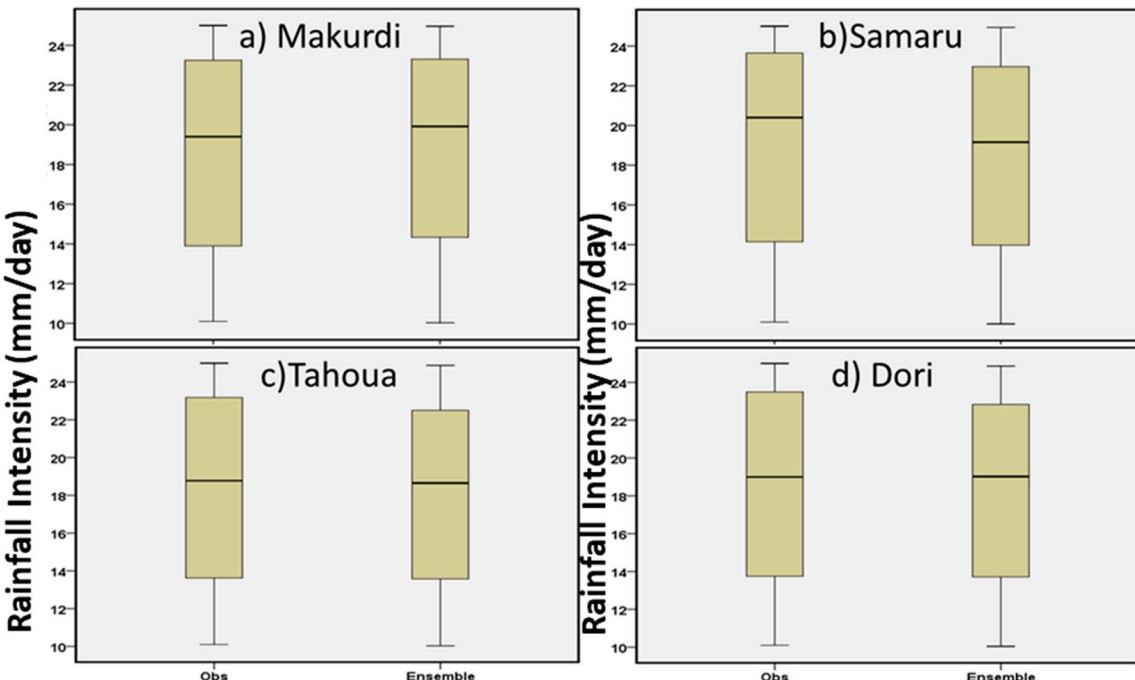

**Figure 4.** Box and whisker plot comparing the CORDEX multi-models ensemble and observed mean daily moderate rainfall intensity (>10 to ≤25 mm) over the (**a**) Southern (Makurdi) and (**b**)Northern (Samaru) Guinea agro-ecological zones and the (**c**,**d**) Sahelian (Tahoua and Dori) agro-ecological zones for the historical period, 1976/1980–2005.

The results may be summarized as follows:

i).     The variance of the ensemble simulated and observational data is statistically significantly different for annual total rainfall, onset, cessation (but not in the Sahel), and duration of season.

ii).    The variance of the ensemble simulated and observed rainfall intensities is similar in the low, moderate and high intensity categories but statistically significantly different in the extreme rainfall category.

The results of the *t*-test for means show that the means of the observational data are not statistically different from the ensemble mean rainfall for all variables and all locations except for the Makurdi (onset) and extreme rainfall intensity (Makurdi and Dori). The boxplots (Figures 3 and 4) qualitatively show comparisons for other statistics between observed and simulated rainfall variables (See Figure 2).

Thus, with few exceptions, these results show that the key statistical parameters based on ensemble simulated intra season rainfall characteristics are similar to observations and, therefore, can be used to analyze the projected rainfall changes in the region.

*3.2. Future Rainfall Characteristics in the Niger River Basin*

3.2.1. Seasonal Rainfall Pattern

Figure 5 and Tables 3 and 4 show the summary statistics and mean seasonal precipitation changes for both RCP4.5 and RCP8.5 emission scenarios for the future period relative to baseline. The results show statistically ($p < 0.05$) significant increases in the future mean annual rainfall for Northern and Sahelian locations for both RCP4.5 and RCP8 but not in the Southern Guinea zone. These results are consistent with other studies (e.g., [26,36,37,39,41], see also scenario 2). However, differences exist even within the same ecological zone. The boxplots show that the observed seasonal rainfall has a much larger range than the projected rainfall, reflecting the well known limitation of climate models to reproduce variability [46,58].

**Table 3.** Change in the seasonal rainfall for the Niger River Basin for the future period (2021/2025–2050) relative to the historical period (1976/1980–2005).

| Agro-Ecological Zone | Location | Average Rainfall (mm) | | | Ensemble Change (%) | | Period |
|---|---|---|---|---|---|---|---|
| | | Obs. | RCP4.5 | RCP8.5 | RCP4.5 | RCP8.5 | |
| Southern Guinea | Makurdi | 1168 | 1219 | 1222 | 4.5 * | 4.4 * | 2025–2050 |
| Northern Guinea | Samaru | 983 | 1086 | 1134 | 10.5 | 15.4 | 2021–2050 |
| | N'Tarla | 826 | 912 | 952 | 12.0 | 26.5 | 2025–2050 |
| Sahelian Zone | Tillabery | 381 | 389 | 392 | 2.0 * | 2.8 * | 2025–2050 |
| | Tahoua | 355 | 399 | 421 | 12.5 | 18.7 | 2021–2050 |
| | Dori | 455 | 501 | 513 | 10.3 | 12.9 | 2021–2050 |

Note, all are statistically significant at 0.05 except * using *t*-test for mean.

**Table 4.** The summary statistics of the intra-season rainfall characteristic in the Niger River Basin for the future period (2021/2025–2050) relative to the historical period (1976/1980–2005). Note, + or − show the direction of change (increase or decrease) for RCP4.5 for the significant events ($p < 0.05$), using *t*-test for mean.

| Agro-Zone | Location | *t*-test for Difference of Means | | | | | | | |
|---|---|---|---|---|---|---|---|---|---|
| | | Annual | Onset | End | Duration | Rainfall Intensity | | | |
| | | | | | | Low | Moderate | Heavy | Extreme |
| **Southern Guinea** | **Makurdi** | 0.258 | 0.001(+) | 0.001(+) | 0.0289(−) | 0.001(+) | 0.001(+) | 0.001(+) | 0.001(+) |
| **Northern Guinea** | **Samaru** | 0.004(+) | 0.048(+) | 0.001(+) | 0.744 | 0.001(+) | 0.002(+) | 0.001(+) | 0.013(+) |
| | **N'Tarla** | 0.003(+) | 0.001(+) | 0.001(+) | 0.001(−) | 0.001(+) | 0.355 | 0.685 | 0.001(+) |
| | **Tahoua** | 0.037(+) | 0.001(+) | 0.001(+) | 0.142 | 0.001(+) | 0.003(+) | 0.054(+) | 0.386 |
| **Sahel** | **Tillabery** | 0.732 | 0.001(+) | 0.653 | 0.003(−) | 0.001(+) | 0.018(+) | 0.014(+) | 0.004(+) |
| | **Dori** | 0.051(+) | 0.001(+) | 0.001(+) | 0.006(−) | 0.001(+) | 0.039(+) | 0.036(+) | 0.041(+) |

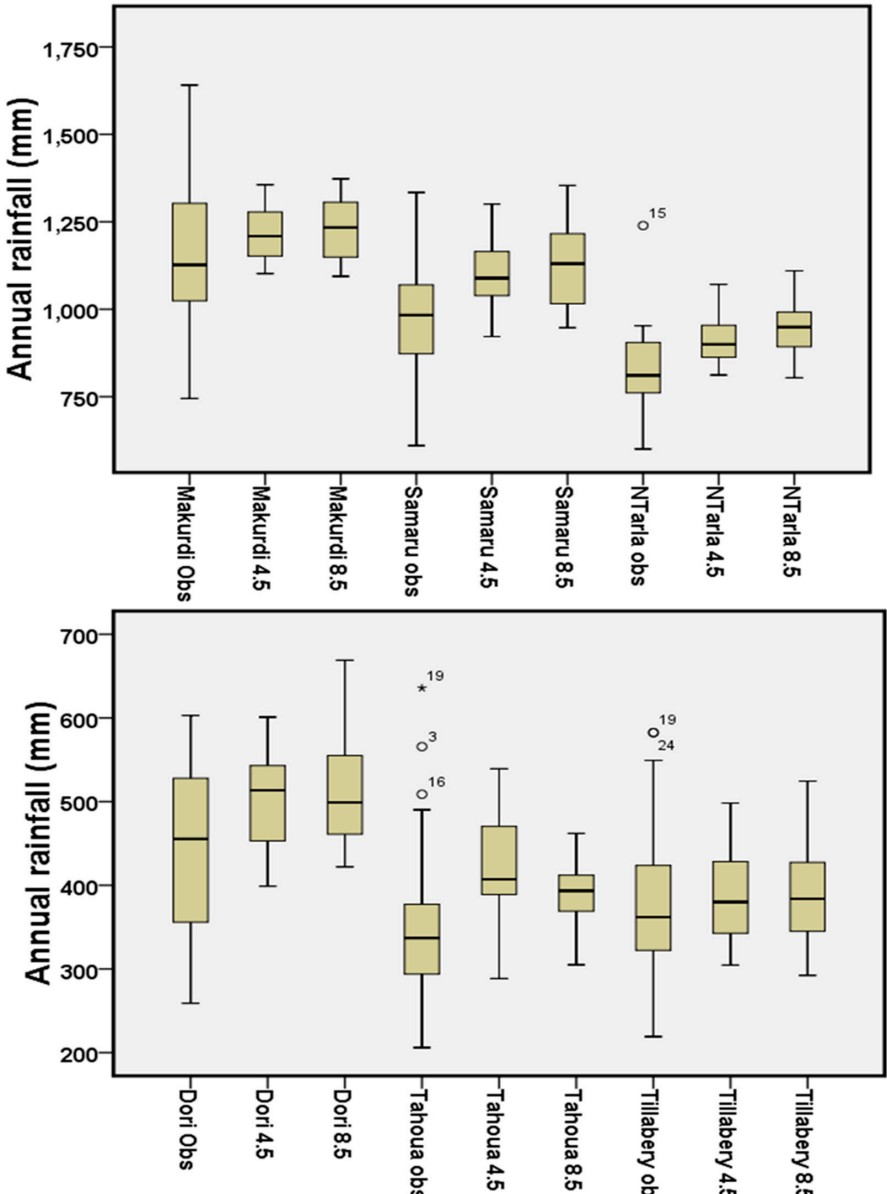

**Figure 5.** The Guinean (Makurdi, Samaru and N'Tarla) and Sahelian (Tahoua, Tillabery and Dori) agro-ecological zones average ensemble annual rainfall changes under the representative concentration pathway (RCP) 4.5 and 8.5 scenarios for the future period, 2021/2025–2050 relative to the baseline period, 1976/1980–2005. Note, all are statistically significant at 0.05.

3.2.2. Projected Change in Intensity and Frequency of Average Daily Rainfall Events in the Niger River Basin

Table 4 compares the future averages of four rainfall intensity categories, relative to baseline. The results reveal a significant change in the means of future rainfall intensities for Southern and Northern Guinea zones and the Sahelian zones for RCP4.5 and RCP8.5 scenarios. However, there is no change in the future rainfall for the moderate and heavy intensities for the N'Tarla location (Northern zone), and extreme rainfall intensity for the Tahoua and Dori locations (Sahel zone). These results are consistent with other studies (e.g., [22,43]) which projected a redistribution of intra-seasonal rainfall for West Africa (See also scenario 4).

Figures 6 and 7 show the expected relative changes of rainfall frequency for different rainfall intensity categories in the Southern (Makurdi) and Northern (Samaru and N'Tarla) Guinea and Sahelian (Tahoua, Tillabery and Dori) agro-ecological zones. The ensemble results agree on the direction of

rainfall change at the low and moderate intensities. The models project that low intensity rainfall is likely to decrease by up to 30% while moderate intensity rainfall will increase by up to 40%. This increase in the moderate rainfall events helps to explain the increase in projected future annual rainfall in the region (see also [17]). The results for higher rainfall intensities are mixed. There is an increase in heavy rainfall but a decrease in extreme rainfall at the Makurdi location at the Southern Guinea zone. For the Northern Guinea zone, there is an increase in the frequency of heavy and extreme rainfall at the Samaru location but a decrease in both heavy and extreme rainfall events at the N'Tarla location. For the Sahelian zone, all locations except Tahoua, show an increase in both heavy and extreme rainfall events.

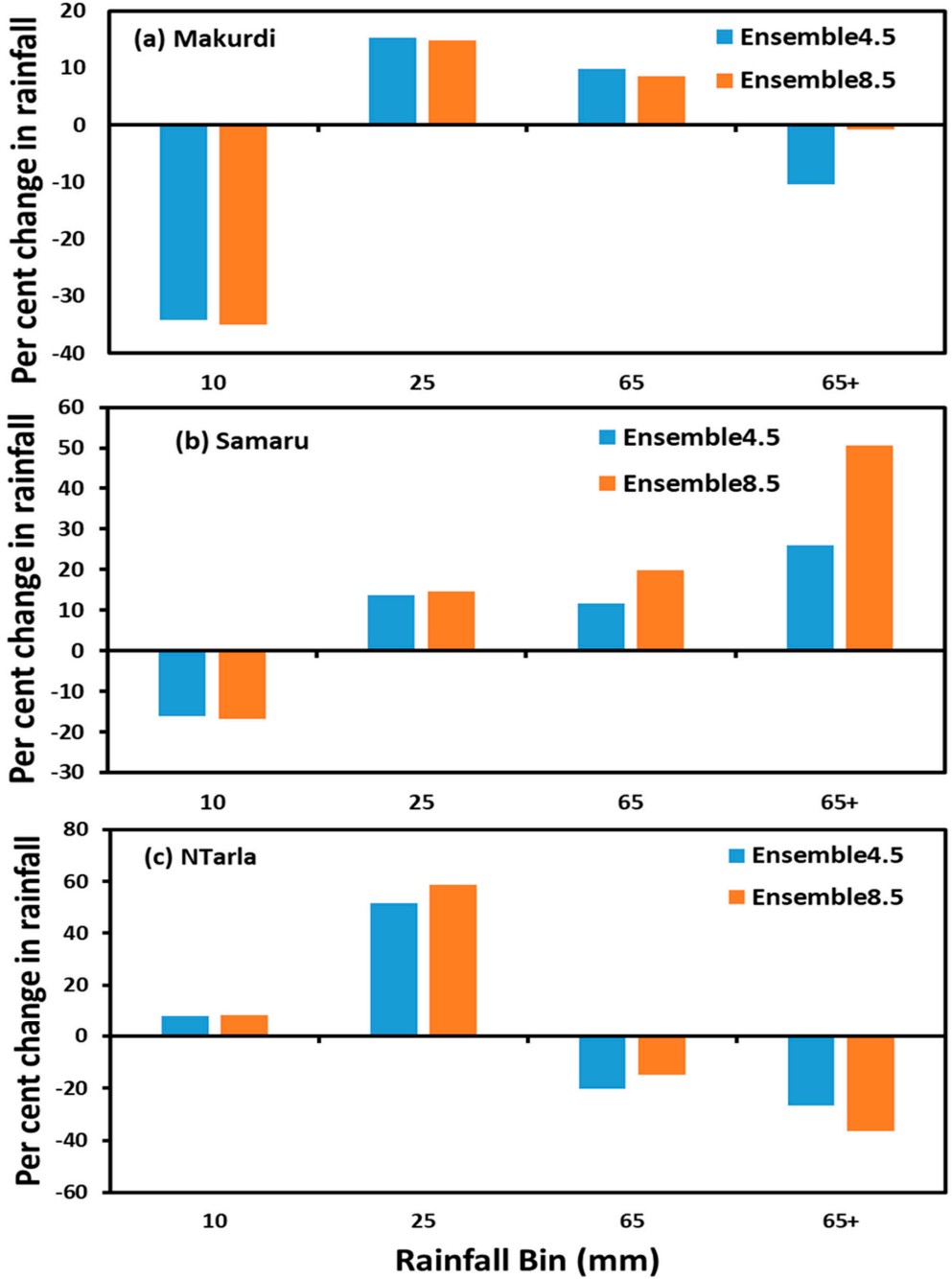

**Figure 6.** (**a**) Southern (Makurdi) and (**b**,**c**) Northern (Samaru and NTarla) Guinea average ensemble change in the frequency of the rainfall events at different intensities for the future (2021/2025–2050) relative to the baseline (1976/1980–2005).

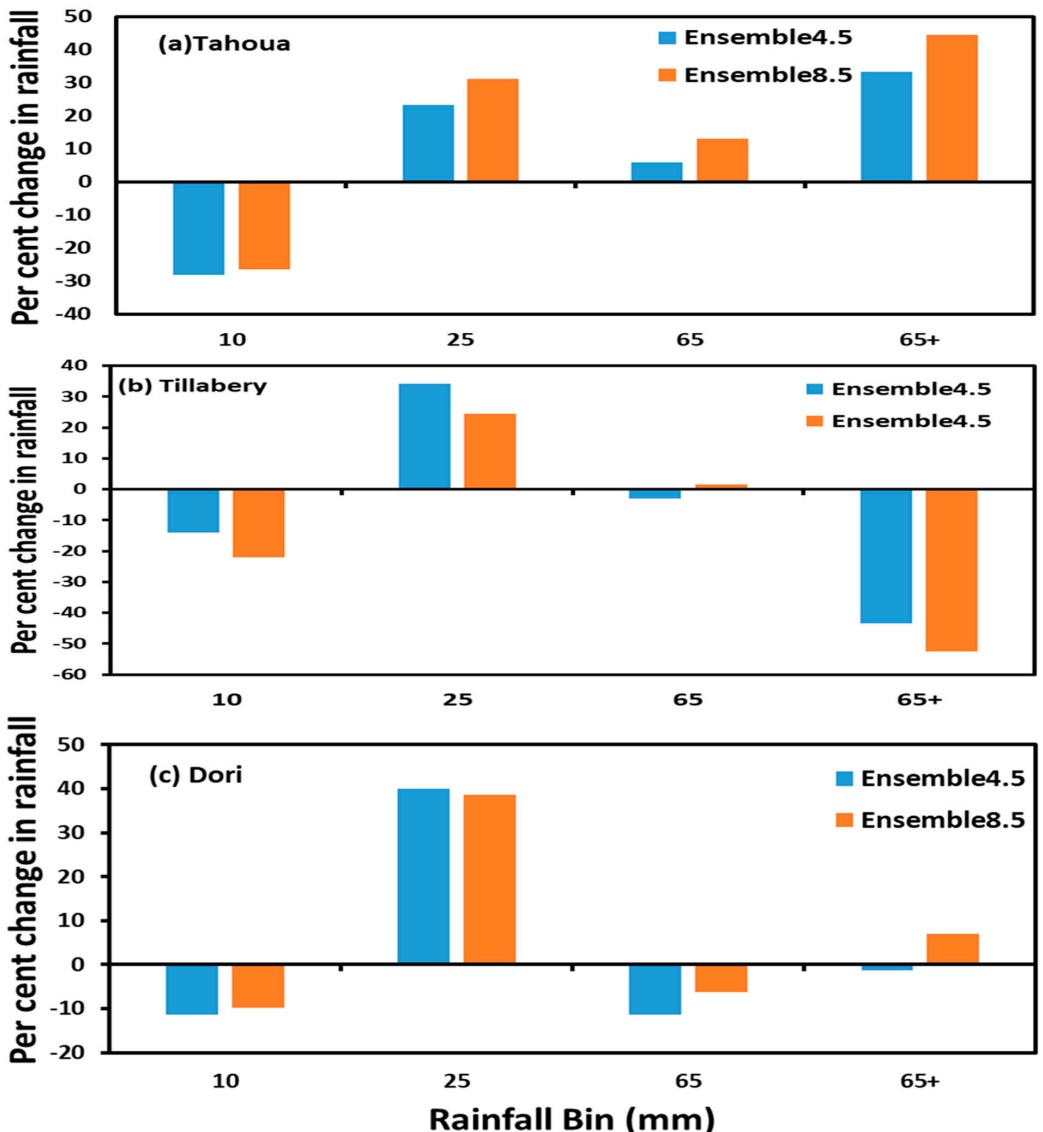

**Figure 7.** Sahelian ((**a**) Tahoua, (**b**) Tillabery, and (**c**) Dori) average ensemble change in the frequency of rainfall at different intensities of rainfall events for the future (2021/2025–2050) relative to the baseline (1976/1980–2005).

The results underscore the previous observation that even within the same zone, climate change can present a unique situation that demands local adaptation policies and practices [36].

*3.3. Onset/Cessation and Duration of the Rainy Seasons in the Niger River Basin*

Figures 8–10 and Tables 4–6 present the summary statistics and results of the mean/earliest/latest dates of onset and cessation of future rains in the Niger River Basin. The results show that on average, the onset of rains will be delayed by between 10 (Tahoua, Dori, Samaru, N'tarla and Makurdi locations) and 32 days at the study locations based on RCP4.5. Similarly, the mean cessation dates of rains will be later by between 10 and 21 days. Despite differences in magnitude, the results are robust and consistent in terms of the direction of change. The overall effect, then, will be a rightward shift of the entire growing season, which will be greater for the start of the rains than for the end. This situation accords with scenario 1 (Table 1). Indeed, as shown in Table 1, this observation is consistent with the findings of several authors (e.g., [16,22,26–31,33]) who have reported similar shifts.

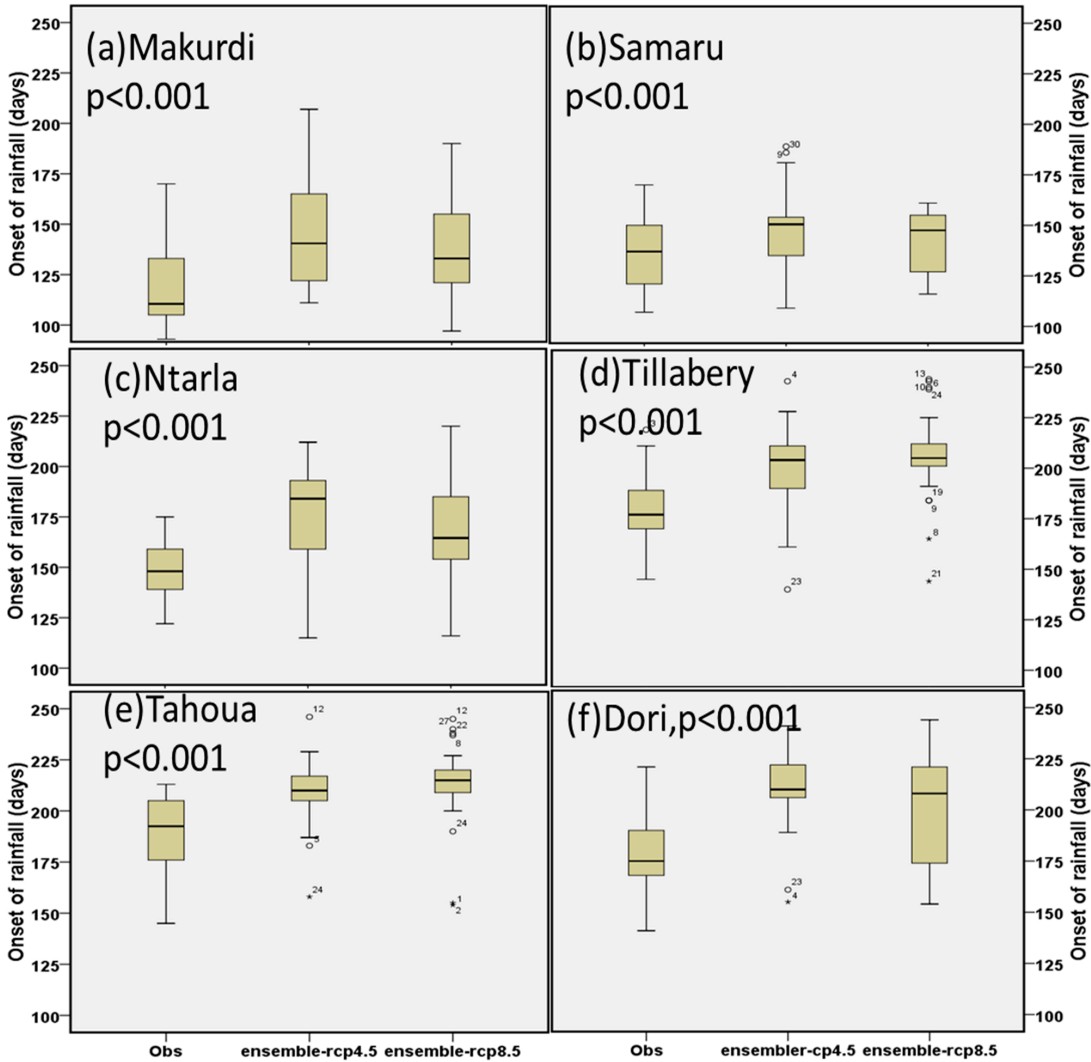

**Figure 8.** (**a**) Southern (Makurdi) and (**b**,**c**)Northern (Samaru and NTarla) Guinea and (**d**–**f**) Sahelian (Tahoua, Tillabery and Dori) average ensemble change in rainfall onset for the future period (2021/2025–2050) relative to the baseline (1976/1980–2005). All in Julian days.

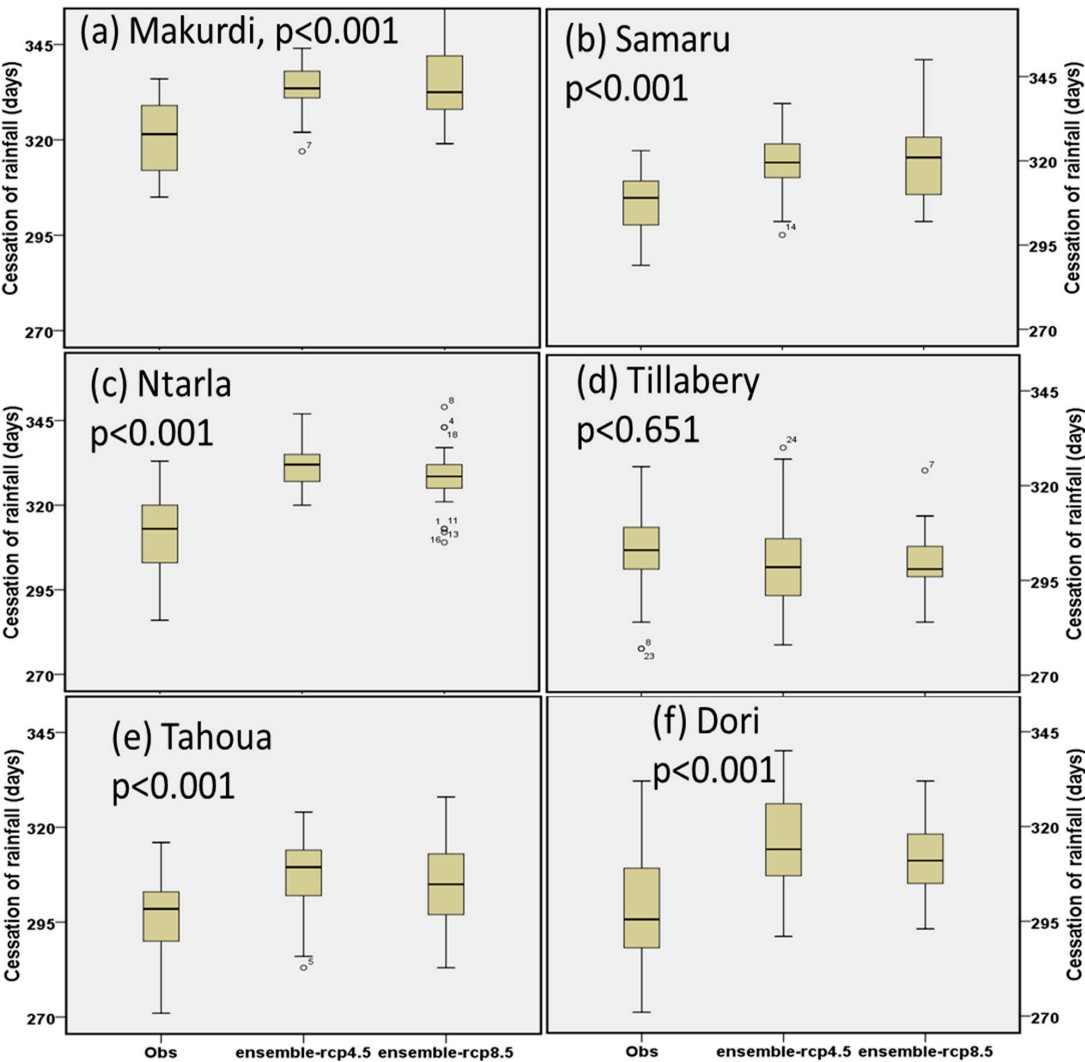

**Figure 9.** (**a**) Southern (Makurdi) and (**b**,**c**) Northern (Samaru and NTarla) Guinea and (**d**–**f**) Sahelian (Tahoua, Tillabery and Dori) average ensemble change in rainfall cessation for the future period (2021/2025–2050) relative to the baseline (1976/1980–2005). All in Julian days.

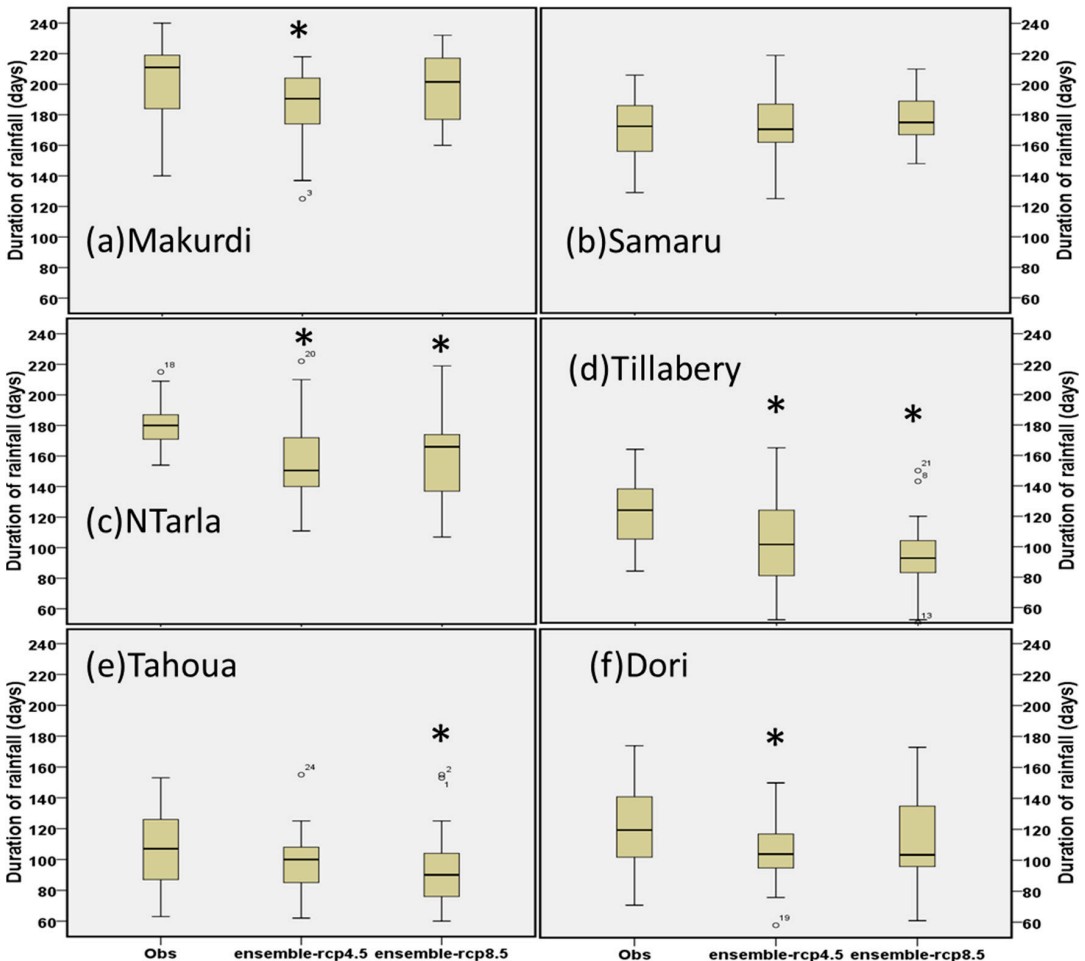

**Figure 10. (a)** Southern (Makurdi) and (**b**,**c**) Northern (Samaru and NTarla) Guinea and (**d**–**f**) Sahelian (Tahoua, Tillabery and Dori) average ensemble change in the duration of the growing season for the future period (2021/2025–2050) relative to the baseline (1976/1980–2005). Note, * indicates a significant change (*p* < 0.05).

**Table 5.** Southern and Northern Guinea average ensemble change in the rainfall characteristic for the future period (2021/2025–2050) relative to the baseline (1976/1980–2005). All in Julian days.

| Southern Guinea (Makurdi) | Mean Onset (Days) | Change (%) | Mean Cessation (Days) | Change (%) | Earliest Onset (Days) | Earliest Cessation (Days) | Latest Onset (Days) | Latest Cessation |
|---|---|---|---|---|---|---|---|---|
| Obs. | 117 | 0 | 320 | 0 | 93 | 264 | 170 | 336 |
| Ensemble 4.5 | 147 | 26 | 334 | 4 | 111 | 317 | 207 | 361 |
| Ensemble 8.5 | 138 | 18 | 335 | 5 | 87 | 319 | 190 | 361 |
| **Northern Guinea (Samaru)** | **Mean Onset (Julian Days)** | **Change (%)** | **Mean Cessation (Julian Days)** | **Change (%)** | **Earliest Onset (Julian Days)** | **Earliest Cessation (Julian Days)** | **Latest Onset (Julian Days)** | **Latest Cessation (Julian Days)** |
| Obs. | 136 | 0 | 307 | 0 | 107 | 289 | 170 | 323 |
| Ensemble 4.5 | 146 | 7 | 319 | 4 | 109 | 298 | 189 | 337 |
| Ensemble 8.5 | 143 | 6 | 320 | 4 | 116 | 302 | 161 | 350 |

**Table 6.** Sahelian average ensemble change in the rainfall characteristic for the future period (2021–2050) relative to the baseline (1976–2005). All in Julian days.

| Sahel (Tahoua) | Mean Onset (Julian Days) | Change (%) | Mean Cessation (Julian Days) | Change (%) | Earliest Onset (Julian Days) | Earliest Cessation (Julian Days) | Latest onset (Julian Days) | Latest Cessation (Julian Days) |
|---|---|---|---|---|---|---|---|---|
| Obs. | 190 | 0 | 297 | 0 | 145 | 271 | 213 | 316 |
| Ensemble 4.5 | 209 | 10 | 307 | 4 | 158 | 283 | 246 | 324 |
| Ensemble 8.5 | 213 | 12 | 306 | 3 | 154 | 283 | 245 | 328 |
| **Sahel (Dori)** | **Mean Onset (Julian Days)** | **Change (%)** | **Mean Cessation (Julian Days)** | **Change (%)** | **Earliest Onset (Julian Days)** | **Earliest Cessation (Julian Days)** | **Latest Onset (Days)** | **Latest Cessation (Julian Days)** |
| Obs. | 178 | 0 | 299 | 0 | 141 | 271 | 221 | 332 |
| Ensemble 4.5 | 210 | 19 | 315 | 6 | 155 | 291 | 241 | 340 |
| Ensemble 8.5 | 201 | 13 | 312 | 4 | 154 | 293 | 244 | 332 |

## 4. Conclusions

In this paper, we used an ensemble of nine bias-corrected GCMs downscaled with one regional climate model to assess change in future rainfall characteristics based on the major agro-ecological zones in the Niger River Basin. The major findings are the following:

1.  For the locations analyzed for this study, the multi-model ensemble means of projected within season rainfall characteristics, including total season rainfall, onset, cessation, duration and rainfall intensities are not statistically different from point scale or site-specific observational rainfall data. However, given significant variability in rainfall over space, as well as uncertainties in future projections, this result must be interpreted cautiously. To investigate future changes in rainfall variables for other locations using a similar approach, it may be necessary to establish or confirm the relationship between site-specific and projected precipitation specific to the study location.

2.  The major changes in future seasonal rainfall include: (i) an increase in mean rainfall of up to 27% in the Guinea zone and about 12% in the Sahelian locations. The result is in agreement with those based on regional models, which also project a larger amount of rainfall increase in the Guinea zone relative to the Sahel [26,39,40]. (ii) The onset of the rainy season will be delayed by between two and four weeks while the mean date of rainfall cessation will also be delayed but by a smaller amount (10–16 days). Thus, the entire rainy season will shift backwards i.e. later in the year. Because the amount of the delay in onset is twice as much as the shift in cessation, the duration of the rainy season will also shorten by about one week. While the absolute value of these changes is small, their potential impacts may be considerably larger due to the variability around these mean dates.

3.  The models project that low-intensity rainfall is likely to decrease by up to 30% while moderate intensity rainfall will increase by up to 40%. However, the results for higher rainfall intensities are mixed. There is an increase for heavy rainfall but a decrease for the extreme rainfall category at some locations. Future studies are needed to investigate the likely impacts of these changes on agricultural productivity and other rainfall-dependent activities.

Despite some caveats, the study demonstrated the value of investigating climate variability at site-specific locations to inform decision-making and climate change management.

**Supplementary Materials:** The following are available online at http://www.mdpi.com/2073-4433/9/12/497/s1, Figure S1: Box and whisker plot comparing the CORDEX multi-models ensemble and observed mean daily heavy rainfall intensity (>25 to ≤65mm) over the Southern (Makurdi) and Northern (Samaru) Guinea agro-ecological zones and the Sahelian (Tahoua and Dori) agro-ecological zones for the historical period, 1976/1980–2005., Figure S2: Box and whisker plot comparing the CORDEX multi-models ensemble and observed mean daily heavy rainfall intensity (>65mm) over the Southern (Makurdi) and Northern (Samaru) Guinea agro-ecological zones and the Sahelian (Tahoua and Dori) agro-ecological zones for the historical period, 1976/1980–2005, Figure S3. Cumulative distribution functions (CDFs) for the N'Tarla station. Minimum daily temperatures are in the top panel while precipitation values are in the bottom. Red CDFs are raw model data, green CDFs are bias corrected and blue CDFs are observed.

**Author Contributions:** U.A. conceived and designed the experiments and analyzed the data; A.T. contributed materials/analysis tools; U.A. wrote the paper and A.T. reviewed the paper.

**Funding:** This research received no external funding.

**Acknowledgments:** We acknowledge the Institute for Agricultural Research, Ahmadu Bello University (I.A.R/ABU) Zaria; Center for Agriculture, Hydrology, Meteorology (AGRHYMET); Agricultural Research Station N'Tarla, Institut D'Economie Rurale (IER), Programme Coton, Station de Recherche Agronomique de N'Tarla, Mali; and CORDEX Africa for providing meteorological data analyzed in this study. The authors are grateful to Claudio Piani (American University of Paris) O. Ogunsola (University of Oklahoma), Mrs Beida Rahama Lele (University of Oklahoma), B. Dogo (Kaduna State University, Nigeria), E. H. Abdou (University of Oklahoma), Teshome Yami (University of Oklahoma), and Omotayo Omosebe (University of Oklahoma) for their valuable input in the data analysis and proof reading of the manuscript.

**Conflicts of Interest:** The authors declare no conflict of interest.

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
