# Peer review of "Projected Changes in Intra-Season Rainfall Characteristics in the Niger River Basin, West Africa"

_atmosphere, doi:10.3390/atmos9120497_

Round 1
Reviewer 1 Report
Numbering of section headings - line number 173 (2.2.0. Methods .....)
Line number 223 - I) -the roman letter should be i) for consistence
Line number 260 - Figure 5 Is illustrating annual rainfall not seasonal rainfall, be specific.
Line number 260 - Table 4 is discussed in section 3.2.2!!! Therefore it should be moved immediately after that section
Line number 321 - Write Tables 5-6 not 4-6
Line number 329 - write: shown in Table1, this observation is consistent.....
Author Response
Dear Reviewer #1,
We are very grateful for the opportunity to revise this paper. We really appreciate you thorough review
We believe the concerns you raised have all been addressed by this version of the paper, which is much improved as a result.
Our point-by-point response to all your comments follows:
Reviewer comments (1)
Numbering of section headings - line number 173 (2.2.0. Methods .....)
Authors response
We have corrected the errors in the numbering to read 2.4, 2.5 and 2.6 respectively.
Reviewer's comment (2)
Line number 223 - I) -the roman letter should be i) for consistence
Author's response
We have corrected this to i)
Reviewer's comment (3)
Line number 260 - Figure 5 Is illustrating annual rainfall not seasonal rainfall, be specific.
Author's response
In this region the term are used interchangeably. Strictly speaking, it is seasonal rainfall. We have added season rainfall in the caption in bracket. The ensemble annual (seasonal) rainfall.
Reviewer's comments (4)
Line number 260 - Table 4 is discussed in section 3.2.2!!! Therefore it should be moved immediately after that section
Author's response
Tables 3 and 4 are first mention in section 3.2.1 and that is why we placed this tables immediately after that section where they are first mentioned.
Reviewer's comments (5)
Line number 321 - Write Tables 5-6 not 4-6
Author's response
Table 4 also present summary statistics of onset, cessation and duration. Therefore, Tables 4-6 is correct.
Reviewer's comments (6)
Line number 329 - write: shown in Table1, this observation is consistent.....
Author's response
We have rewritten to read: Shown in Table 1, this observation is consistent with the findings of several authors
Reviewer 2 Report
Studies in climate change predict an increase in rainfall extreme events associated with increased risk of drought and rainfall variability at different time scales (i.e., intra-seasonal, seasonal, and interannual rainfall variability). These changes in rainfall regime have important implications on ecological, ecohydrological and economic aspects. This study by Uvirkaa Akumagaa and Aondover Tarhuleb used nine ensemble bias-corrected climate model projection results under RCP4.5 and RCP8.5 to investigate the projected changes in intra-seasonal rainfall characteristics in the Niger River Basin, West Africa. They found a general increase in rainfall in these locations with increased occurrence of moderate-heavy rainfall events. Moreover, it shows the increase in a shift in the future onset/cessation and a decline in duration of the rainy season in the Basin. They also discuss the implications of changes in rainfall regime on food production, et al.
Here I have some suggestions to improve this study. First, as concluded (see conclusion 1), the study suggests that results in ensemble mean of models are not statistically different from point scale or site specific observational rainfall data and may therefore be used to investigate future changes. While this is a reasonable hypothesis in particular for temperature, it would be cautious that rainfall projections in models (even in ensemble mean) have large uncertainty. This point should be clarified in the discussion.
Second, to improve the implications of this study, the discussion should add more contents about the projected changes in rainfall regime on vegetation dynamics et al. At this time, it only briefly mentioned about foot security. In fact, there are extensive studies which investigate vegetation greening (Hickler et al., 2005) in Sahel, how changes in rainfall regime affect vegetation, dust emission and its feedbacks to climate in Sahel (Yu et al., 2015).
I also have some minor suggestions to improve this study.
Line 47: what does “p.66” refer to?
Line 56: delete “also”
Line 57-58: may consider moving “for other activities…” after “far more critical”; may consider deleting “overall quality”
Lines 61-64: may consider rephrasing them as an independent sentence.
Line 93-94: may consider rephrasing this sentence
Line 160: change “have skill” to “be good”
Lines 163-165: it would need to add more details about how to conduct the bias-correction process, which is key of this study
Figure 2: y axis is rainfall frequency. In hydrology, rainfall frequency is typically expressed as number of rainfall events per day, or pre year. Here it showed the total rainfall events (frequency) over the whole 1976/80-2005? Thus, it needs to better define rainfall frequency. Also usually t test should have error bar?
Figure 3: y axis is rainfall intensity rather than rainfall. It would be better to also clarify that it is box and whisker plot, which show minimum, first quartile, median, third quartile, and maximum.
Figure 6: need to better define y axis. Is it rainfall frequency change?
Figure 8: it is surprising that P = 0.00.
In the section of results and discussion: at this time, it reads more like results. It would be better to add some discussion about the implications of changes in rainfall regimes on biocrust (i.e., Jia et al., 2018), vegetation dynamics, or biogeochemical cycling et al.
Line 365: delete “;”
References
Hickler et al., 2005. Precipitation controls Sahel greening trend
Yu et al., 2015. Dust‐rainfall feedback in West African Sahel
Jia et al., 2018. High rainfall frequency promotes the dominance of biocrust under low annual rainfall

Author Response
See the attached for our response
